# Indifferent, ambiguous, or proactive? Young men's discourses on health service utilization for *Chlamydia trachomatis* detection in Stockholm, Sweden: A qualitative study

Frida M. Larsson[1]*, Anna Nielsen[1], Erica Briones-Vozmediano[2,3], Johanna Stjärnfeldt[1], Mariano Salazar[1]

**1** Department of Global Public Health, Karolinska Institute, Stockholm, Sweden, **2** Department of Nursing and Physiotherapy, University of Lleida, Lleida, Spain, **3** Healthcare research group (GRECS), Biomedical Research Institute (IRB), Lleida, Spain

* frida.larsson@ki.se

**Data Availability Statement:** We follow the EU General Data Protection Regulation (2016/679)

## Abstract

### Introduction

*Chlamydia trachomatis (C. trachomatis)* infection is the most commonly reported sexually transmitted infection in Sweden and globally. *C. trachomatis* is often asymptomatic and if left untreated, could cause severe reproductive health issues. In Sweden, men test for *C. trachomatis* to a lesser extent than women.

### Aim

To explore factors facilitating and constraining Swedish young men's health care utilization for *C. trachomatis* detection and treatment.

### Method

A qualitative situational analysis study including data from 18 semi-structured interviews with men (21–30 years). Data collection took place in Stockholm County during 2018. A situational map was constructed to articulate the positions taken in the data within two continuums of variation representing men's risk perception and strategies to test for *C. trachomatis*.

### Results

Based on the informants' risk perception, strategies adopted to test and the role of social support, three different discourses and behaviours towards *C. trachomatis* testing were identified ranging from a) being *indifferent* about *C. trachomatis* -not testing, b) being *ambivalent* towards testing, to c) being *proactive* and testing regularly to assure disease free status. Several factors influenced young men's health care utilization for *C. trachomatis* detection, where the role of health services and the social support emerged as important factors to facilitate *C. trachomatis* testing for young men. In addition, endorsing traditional

regarding restricted access to the data. The sensitive nature of our qualitative data means that even if the names and other identifying information (address, telephone number, etc.) of the young men part of this research have been removed from the transcripts, there is a possibility that they could be identified thorough their narratives (as they share personal experiences). Excerpts of the data transcripts that are relevant for the study are available within the paper in the result section. Requests for access to more data should be made to the Research Data Office at Karolinska Institutet via rdo@ki.se and if permitted by law and ethical approval, decided on a case by case basis, the data can shared.

**Funding:** MS got financial support from FORTE – Swedish Research Council for health Working Life and Welfare (2016-00594) to conduct this research project (URL: forte.se). EBV received financial support from the mobility programme Jose Castillejo 2019 (Spanish Ministry of Education), the Serra-Hunter University Programme of the Generalitat de Catalunya, and the University of Lleida Research Promotion Aid. The funders had no role in study design, data collection and analysis, decision to publish, or preparation of the manuscript.

**Competing interests:** The authors have declared that no competing interests exist.

masculinity domains such as leaning on self-reliance, beliefs on invulnerability and framing men as more carefree with their sexual health than women delayed or hindered testing.

## Conclusion

Testing must be promoted among those young men with indifferent or ambivalent discourses. Health systems aiming to increase testing among those at risk should take into consideration the positive role that young men's social support have, especially the level of social support coming from their peers. Additionally, endorsement of traditional masculinity values may delay or hinder testing.

## Introduction

*Chlamydia trachomatis (C. trachomatis)* is the most commonly reported sexually transmitted infection (STI) globally, in Europe and in Sweden [1,2]. WHO estimated a global incidence of 127 million new cases per year (2009–2016) [3,4]. Global prevalence is higher for women (3.8%) than men (2.7%) in the age group 15 to 49 [3].

In Sweden, in 2019, women accounted for 56% of the reported new cases [2], however the estimated prevalence of *C. trachomatis* in Swedish men (9.0%) was higher than in women (4.4%) [5]. The highest infection rate in Sweden was found in Stockholm County (2019) with 417 cases per 100,000 inhabitants as well as the highest number of reported cases in the country (9,930) [2].

Available data shows that the *C. trachomatis* epidemic in Sweden is focused on young people [2]. Youth aged 15–29 years accounted for 78.7% in 2019 [2]. Acquiring *C. trachomatis* has been reported to be associated with high risk-behaviours including alcohol consumption, multiple sexual partners [6], and lower education [7]. Heterosexual transmission was the most common route of transmission in Sweden (94% and 83% for women and men respectively) [2]. The asymptomatic nature of *C. trachomatis* [8,9], can potentially cause serious health consequences such as infertility. Men might experience discharge from the urethra, burning sensation/pain during urination, pain in the testis, and/or chronic prostatitis [10].

### Swedish health care system for *C. trachomatis* detection

Sweden has a so-called opportunistic screening approach for *C. trachomatis* [11], which includes offering testing to one/more than one specific group of asymptomatic people (e.g. pregnant women or youth), and case finding through contact tracing [11]. In Sweden, testing for *C. trachomatis* is free of charge for the individual. Testing can be done at Youth Health Clinics, primary health care facilities, STI-clinics, and on-line/self-testing [12].

At the Youth Health Clinics in Stockholm County, 30% of *C. trachomatis* tests were conducted among men [13]. Furthermore, the incidence rate for *C. trachomatis* was almost two times higher for men than women (11% vs. 5.7%) [13]. The differences in incidence rate could be explained by women more commonly being tested without symptoms within the opportunistic screening program, whereas men are usually tested when having symptoms [5]. In evaluations it has been shown that the differences in testing proportion between sexes are similar within Internet-based home testing for *C. trachomatis*, i.e. men test more seldom [14]. The combination, men testing less than women, and *C. trachomatis* often being asymptomatic, contribute to the epidemic in Sweden, i.e. men are unaware of their *C. trachomatis* status and

consequently infects others. In order to stop the *C. trachomatis* epidemic, it is important to increase young men's access to *C. trachomatis* testing and treatment services [15].

## Health seeking behaviour for men

It is known that men generally use health care services less extensively than women which in turn lead to poorer health outcomes [16]. Delayed health seeking is the focus in a growing body gender-specific literature, where 'traditional masculine behaviour´ has been described as a reason behind the delay [17].

Barriers to sexual health care clinics has been related, by men, to the clinics being a place for women´ [18]. Furthermore, it has previously been described that *C. trachomatis* is not considered a severe infection [19]. Perceived severity of suspected condition will influence the health seeking behaviour [20]. Among men with STI-related symptoms such as urethral discharge or dysuria, data indicate that more than one in four delayed seeking care for more than a week. At the same time, this group of men were more likely to continue to engage in unsafe sexual practices [21].

While barriers to testing among women are rather well studied [6,22], (e.g. ignorance, inaccurate information, denial, stigma, confidentiality, and privacy concerns), the barriers and facilitators for men are less known. To decrease the pool of untreated *C. trachomatis* infections among men, it is vital to improve understanding of what factors influences young men´s health service utilization for *C. trachomatis* detection and treatment. To address this gap, this qualitative study aimed to explore factors facilitating and constraining Swedish young men's health care utilization for *C. trachomatis* detection and treatment.

## Theoretical framework

The theoretical framework guiding this paper was the Andersen´s Model of Health Service Utilization [23]. In the model, usage of health services is determined by three dynamics: *predisposing factors*, *enabling factors*, *and need*. Predisposing factors include characteristics such as gender, age, ethnicity, and health beliefs. Enabling factors include family support, income, and accessibility to health care in the community. Need represents both perceived and actual need for health care services [20].

The model describes a number of interacting factors influencing health service utilization. These factors are: *1. The Environment* (health care system and external environment such as physical, political, and economic environment), *2. Population characteristics* as mentioned above (predisposing factors, enabling factors and need). *3. Health Behaviour* (personal health practices and use of the health care system) and *4. Outcome* (perceived health status, evaluated health status and customer satisfaction) [20].

## Materials and methods

### Study design and setting

The study is based on a qualitative research design including data from in in-depth interviews. The study took place in Stockholm, the capital of Sweden [24]. In 2019, total population in Stockholm County was approximately 2,38 million and within the population about 159 000 were young men between 20–29 years of age [25]. The city of Stockholm includes thirteen districts and the mean income and education level vary widely between the district areas [26].

## Participants and data collection

Researchers applied purposive sampling technique using the inclusion criteria: young men (20–29 years) living in Stockholm county, regardless of experiencing *C. trachomatis* testing or not. Participants were recruited through adds posted in public places (such as metro, educational institutions, and libraries) and by using snowball technique [27]. The recruitment strategy also involved visiting clinics and schools to informing about the study. Interested participants received information about the study and any questions were answered to by the research team. Those agreeing to participate were scheduled for in-depth interviews.

In total, 18 in-depth interviews were collected between February 2018 to February 2019. The participants´ ages ranged from 21 to 30 years and residence in different socioeconomic districts of Stockholm. Most reported upper secondary school as their highest level of education (14 of 18 young men), eight participants were employed, nine were students and one was unemployed. Half of the participants reported that they were in a relationship, and 13/18 had previous experience of STI testing.

## Data collection

A semi-structured interview guide with open ended questions was used. The guide was designed based upon a literature review on the topic and the researchers' previous experience on the field. The topics explored in the guide included: experiences of seeking health care in general and for STIs, understanding of sexual risk-taking behaviours, access to and quality of social support, and understanding of masculinities. In addition, some questions in the interview guide were formulated as hypothetical questions and vignettes. The technique of using vignettes enable young people to engage, explore and identify sensitive topics [28]. Follow-up and probing questions were asked to explore and gain deeper knowledge of the topics under study [29].

The data was collected by two trained researchers (FL and JS) and preceded by a pilot interview. The location of the interviews was decided by the participant. The interviews were conducted in a private room at Karolinska Institute, at a healthcare clinic, or via video-calls. The interviews were conducted in Swedish or English, lasted between one and two hours and were recorded using an audio device. Throughout the data collection process, field notes were taken, and emerging ideas were discussed with the research team. Data was collected until saturation was reached and the interviews were transcribed verbatim [29].

## Data analysis

A constructivist situational approach was used to analyse the data [30]. This method allowed us to identify the different discourses that the participants´ used to test for *C. trachomatis* and other STIs and to map how these discourses varied and intersected along different axes of variation.

The analytical process started by reading the transcript to get familiar with the data. Afterwards, inductive open coding was used to identify concepts emerging from the data by using Open Code 4.03 [31]. Similar codes were then merged into categories. The relationships between the categories were identified and used as a starting point to identify the overall discourses that men used in this setting for health care utilization for *C. trachomatis* and STI testing.

We identified three different discourses, which are "metaphorical constructs grounded in empirical data capturing behaviours and discourses" [32]. Finally, a situational map was created, showing the differences and similarities in the positions that young men took on *C. trachomatis* and STI and testing.

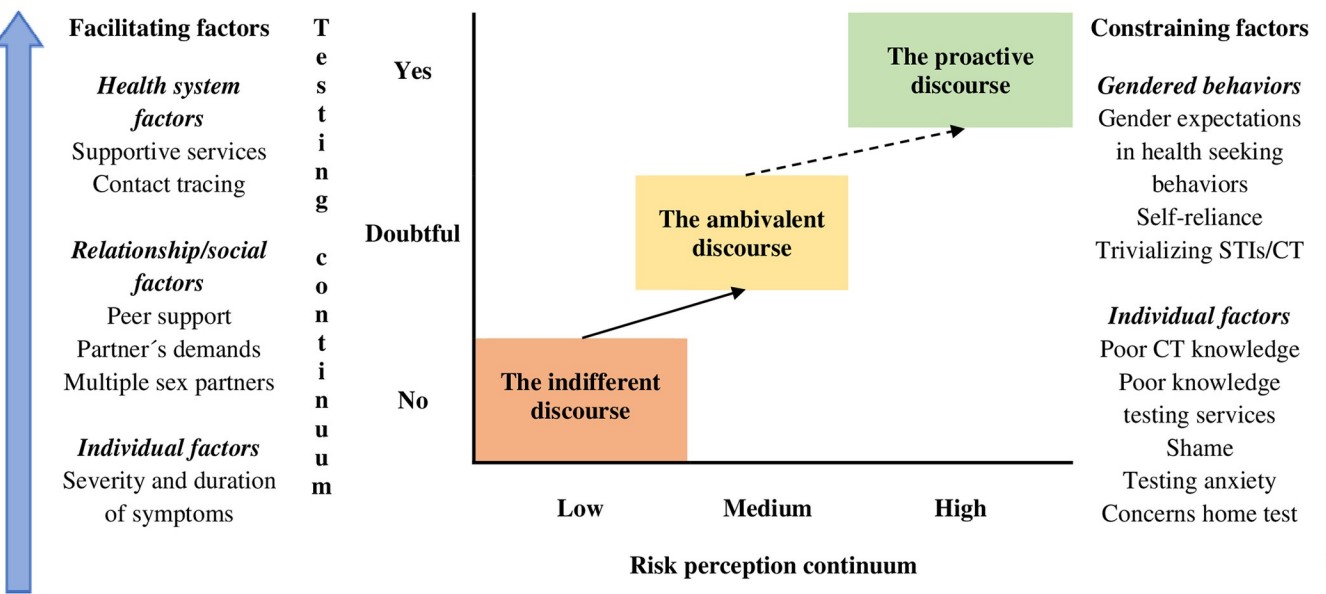

**Fig 1. Young men's discourses related to *C. trachomatis* testing.** Positional map over how Swedish you men´s discourses overlap and differ within two continuums of variation: Own risk perception of getting infected with CT and attitude toward testing continuums.

## Ethical considerations

The study protocol was evaluated and approved by the Regional Ethics Review Board in Stockholm (2017-06-29). Participants signed a written informed consent before participating in the study. Each participant got information prior the interviews about the study aim and its sensitive topic, that participation is voluntarily, that they can withdraw at any time, how their anonymity will be protected, and confidentiality assured. They also gave consent to be tape recorded and was informed that the transcribed material would not be containing any personal details. All materials related to the study were downloaded and stored on a password secured computer. Participants´ received a stipend of movie tickets after participating as a token of appreciation.

## Results

We identified three different discourses: the indifferent discourse, the ambivalent discourse, and the proactive discourse, representing how Swedish young men understand and negotiate health services utilization for *C. trachomatis* testing. We used a positional map to show how these discourses overlap and differ within two continuums of variation: own risk perception of getting infected with *C. trachomatis* and attitude toward testing continuums (Fig 1). The arrows in the positional map show how these discourses can evolve depending on individual, relationship and health system facilitating and constraining factors. In the following, we describe in detail these discourses and the factors facilitating or constraining *C. trachomatis* testing.

### The indifferent discourse: *"C. trachomatis is not a big deal, why should I care?"*

This discursive position is characterized by a negative attitude towards testing and a low perception of an individual risk for *C. trachomatis* infection. Knowledge on the disease´s symptoms, treatment and consequences were scarce, and presence of symptoms were a crucial part

in getting tested for chlamydia. Thus, *C. trachomatis* was constructed as a mild curable disease when compared other STIs such as HIV.

> P: *"I am not afraid of anything (any STI) except from the one that is not curable, HIV or aids or what it is called, it's the biggest fear that exists, I think it's the same for everyone, I do not think I am alone about that eeh"*
>
> – (Participant 1, 28 years)

Within this discourse, knowledge on different testing services availability was limited; especially when identifying testing services available other than the youth health clinics that are available only to those 23 years and below. One informant commented:

> P: *"I actually wanted to test (for STIs including chlamydia infection) a few weeks ago, just to be safe eh but I didn´t do it, . . . it didn´t feel very easily accessible, it doesn´t, maybe it is but I think more information is needed since people feel like this, before one could go to the youth clinic, it was smooth, but when you passes that limit, when you are a young adult, it´s not the same, you have to grab it yourself, walk somewhere yourself, there (at the youth clinic), it was more like a common place for all. . ."*
>
> – (Participant 3, 24 years)

The indifferent discourse involved more concern about pregnancy prevention than protecting against STIs. Even if condom had not been used during sex and there was a concern about STIs, testing was not prioritized (and sometimes expressed as hindrance) as *C. trachomatis* was perceived as an innocuous disease. Trust in partner was, according to the indifferent discourse, considered to reduce the need to test for chlamydia even if protection had not been used with a new partner. Concerns about the test procedure and anxiety related to a positive test and its consequences to men´s sexuality (i.e., the need to abstain from sex while being treated) were also expressed in this discursive position.

> P: *". . . Many more are worried about making someone pregnant, if a girl says that she is eating birth control pills or something, then you are satisfied, you don´t think it can, you can get a STI. . ."*
>
> – (Participant 18, 28 years)

## The ambivalent discourse: *"More symptoms force you out of denial"*

This discursive position overlaps with the previous one. For example, *C. trachomatis* infection was also constructed as a mild easily curable disease, pregnancy prevention was more important than using condoms to protect against STIs and testing for *C. trachomatis* was not prioritized. The ambivalent discursive position can be described as an evolution from the previous one and a key difference was its attitudinal change towards testing from a negative to ambiguous stance fostered by a low to medium risk perception of acquiring *C. trachomatis*.

> P: *"Yes, I think that, since it´s very easy (to test for C. trachomatis), one should probably do the test because yes, it takes no time, but, if I would have been with several people then I think I would have gotten a bit more, yes now it's time, why not do the test"*
>
> – (Participant 12, 24 years)

This discourse hesitance towards testing was constructed around the informants' shame of having a STI or needing to show their genitalia during a medical examination, uncertainty if discomfort in genital area is due to a STI, fears of being judged by the staff or other peers, fear of having a severe infection and concerns related to the test procedure (using a swab to collect the test). For example, one informant was anxious about testing needing to introduce *"a long stick into the urinary track "*– (Participant 18, 28 years).

Despite its hesitant attitude towards testing, the act of testing itself was constructed as a function of individual factors, relationship factors, and system factors. Increased severity and duration of genital symptoms were described as key individual factors promoting testing. External factors also emerged as key facilitators to overcome this discourse hesitance towards testing. At the relationship level, the perception that the partner had multiples sexual contacts, as well as the men´s partners demanding that they get tested were keys factors facilitating testing. Entering a new steady relationship was also a key factor for the ambiguous discourse to get tested as they do not want to infect their partner. Finally, receiving a contact tracing letter from the health authority was a key determinant for this discourse to overcome testing ambiguity as well as for the indifferent discourse to test.

> P: *"Yes well, I got, it was a long time ago when I was 18, I got a letter sent home that they had, someone, a partner had, a sexual partner had tested positive for chlamydia and then you should inform about all (sexual partners) you have been with the last 6 months and then she stated me, I went and got tested but it was negative anyway"*
>
> – (Participant 4, 27 years)

## The proactive discourse: *"If I have the chance to test—I´ll do it"*

This discursive position was characterized by a high-risk perception of acquiring *C. trachomatis* and a positive attitude towards testing. *C. trachomatis* was constructed as a disease that needs to be taken seriously due to risk of complication such as infertility for both sexes. Knowledge on *C. trachomatis* symptoms, treatment and consequences were high. For example, the recognition of asymptomatic infection, possible oral and rectal *C. trachomatis* transmission and the treating *C. trachomatis* infection with antibiotics.

> P: *"I know it is the most common sexual transmitting disease ehm I know a little about the symptoms eh well, mainly that its hurting when you go to the toilet but otherwise. . . it´s also common to not have any symptoms at all when you have it, eh yes it is this notification obligation, that you need to report if you have it. . .and then you get a letter home and you need to get tested, it (chlamydia infection) is included in the communicable disease act. . . (I know) you eat antibiotics, 10 days"*
>
> – (Participant 18, 28 years)

A key characteristic of this discourse was the lack of hesitance towards testing and its high knowledge of the different services available for testing. Testing was framed as an act of responsibility towards themselves or others and a part of a healthy life. A difference from the previous discourse was that testing was normalized and not perceived as shameful or anxiety inducing.

> P: *. . . No, I perceive that it (STIs and testing) can be that (shameful), I do not think so myself eh I notice it can be but, but I am working pretty much, I have taken a stand myself to try to*

*normalize it as much as possible eh so I do not have any problem to talk about it (STIs and testing) . . ."*

– (Participant 5, 29 years)

To this end, testing was not only a function of the presence and/or severity of symptoms but proactive protective strategy even if uncertain about own infection or exposure. For example, testing was performed as a precautionary measure to prevent the spread of the disease when having unprotected sex with a new partner, when changing sexual partners or having multiple sex partners, when committing oneself to a long-term monogamous relationship or even to confirm that one is not an asymptomatic carrier.

P: *". . . I had been single for a while and then it felt natural (to test), why not be sure one is healthy, I didn´t have any suspicion about it (being infected), then it was also for the sake of my (new) girlfriend, she thought it was a good idea, I also did it to calm her, but it was nice to get the answer even if I didn´t suspect anything"*

– (Participant 4, 27 years)

## The role of social support and health system experience as testing facilitators

Social support was a key factor to overcome the informants´ negative or ambiguous attitudes towards testing. The main source of support were other trusted male peers. Support facilitated testing by different pathways. Having the opportunity to talk to a friend allowed the informants to release their anxiety over testing, to recognize that the problem was solvable and to received advice and encouragement/pressure on how and where to test. Other pathway that facilitated testing was the opportunity to visit the testing facility with a trusted peer to help them overcome any fears on the procedure.

P: *"They (friends) would probably be understandable but say*; *go and get tested, yes most of them would say that they also will go and get tested. . . it feels like its always like that, someone says that they will get tested and another always like*; *fuck lets go and get tested together, I believe that people think it's difficult to get tested alone, then you go in a group as when you were young. . . then you are pushing each other to book an appointment and go. . . it is nice, it's like you understand what you need, that you need support and a small push"*

– (Participant 17, 28 years)

Some informants did, however, not feel comfortable talking to friends about STIs. The topic was perceived as private and a subject that one keeps to oneself due to the negative societal views. Playful jargons and unserious reactions from peers were other hinders for talking to friends. It was expressed that peers often jokes with each other and not least about STIs. They could for example say: *"haha what the fuck have you done, who have you been with haha"*– (Participant 7, 25 years). However, some of the informants expressed that they still talk to their friends despite these comments since they know it´s just a joke.

For those who had tested, the overall experience was positive. Health services were described as friendly, fast and the testing procedure easy to follow. Choice of health clinic to visit was based on proximity, convenience, and drop-in hours. About half of the participants was aware of the home-test and some expressed concerns on home-testing for *C. trachomatis.*

For example, worries about performing the test correctly and the test being lost in the mail were factors influencing the type of test chosen (health facility vs. home test).

## Gendered expectations, masculinities, and testing for *C. trachomatis*

Across all discourses, women were constructed as being wiser, more careful, thoughtful, and responsible for their sexual and reproductive health than men. This depiction of women was linked to men framing women as having more *C. trachomatis* symptoms and being more at risk for severe STIs/ *C. trachomatis* complications than men. This construction was used to explain why women needed to test and tested more for STIs/ *C. trachomatis* than men.

> P: *"The symptoms (for chlamydia infection) in women might be more visible, yes they, they simply appear a little more than in men, and then it might be because men are more lazy when it comes to their own genital health and women are more careful about what is going on down there (in genital parts), and they do regular check-ups at the gynaecologist and such, I would say that it only has to do with that, that women are more aware about what can happen down there and men are more nonchalant when it comes to sexual transmitting diseases"*
>
> – (Participant 4, 27 years)

In addition, our informants expressed that gendered differences in testing were due to a more supportive societal environment that normalized testing for STI/ *C. trachomatis*. At the family level, they discussed how they perceived women to have a more open communication regarding sexual health including testing with their parents than men, which provided them with more support and knowledge on STIs. At the societal level, the informants discussed how the health system created more opportunities for women to test since they had regular gynaecological check-up and specialized staff to care for their sexual health needs. Finally, the informants expressed that testing becomes more natural for women since they early on need to think about their sexual health due to menstruation and other genital problems.

> P: *"I will try not to be prejudice but I think women have better contact with the health care, it can be that eeh it starts with the menstruation, then I think it's something parents are prepared for and they are there for their daughter, it becomes a more common thing, while men don´t have that kind of body so I think girls and women tend to have support already from the beginning, not all have it, but guidance through it, eeh I think it´s more macho culture among men so I don´t think it's a topic of conversation as often, however, it can be so that women talk about it in a different way, more mature way since its more common for them I think eh yes"*
>
> – (Participant 3, 24 years)

The discourses identified in this paper did not perceive STI/ *C. trachomatis* testing as unmanly. However, endorsement of traditional masculinity domains such as sexual risk taking, and perception of invulnerability were present mainly in the indifferent discourse and to a lesser degree in the ambiguous discourse. Testing was not considered a priority since the risk of contracting STIs while having sex without condoms was downplayed or disregarded.

> I: *"Why do you think you never tested for it (C. trachomatis)?"* P: *"You think you are strong, nothing can affect you, so, that is what I think"*
>
> – (Participant 13, 27 years)

In the ambiguous discourse, endorsement of self-reliance (reluctance to seek help) was another feature of traditional masculinity that contributed to this discourse hesitance to test since it limited the beneficial effect that accessing social support had on testing. One informant commented: "*STI/ C. trachomatis is something that one had to solve by himself*"– (Participant 14, 29 years).

## Discussion

We sought to explore factors facilitating and constraining Swedish young men's health care utilization for *C. trachomatis* detection and treatment, and we found three different discourses related to men's *C. trachomatis* testing: the indifferent discourse—inactive in relation to testing, the ambivalent discourse–*test if* needed or forced, and the proactive discourse—active in relation to testing.

In line with Andersen´s Model of Health Service Utilization [23], describing the enabling factors and needs, we found several factors affecting the discourses and consequently the health seeking behaviour. Men in our study described testing for STIs after having multiple partners and unprotected sex (ambivalent and proactive discourse), which was also reported in a recent New Zealand study [33]. Similarly, a Swedish study exploring sexual risk taking among men testing for STI, found that testers had engaged in unprotected sex during the past year at 3.8 (median) occasions, and that more than one third had used alcohol at their last sexual encounter [34]. As described by Lyons [35], testing positive for STI was associated with casual sex and greater number of partner. It seems thus as young men are aware of the danger of sexual risk-taking which in turn leads to testing according to the proactive and occasionally the ambivalent discourse. Nevertheless, the indifferent discourse expressed unwillingness to test regardless exposure to risk.

Not surprisingly, men in our study revealed testing was associated with the presence of persisting symptoms and their notion of severity of *C. trachomatis* sequelae. More so, a contact tracing letter was also a strong predictor of testing for *C. trachomatis*. The combination of *C. trachomatis* screening and contact tracing has been found to be successful for finding cases and halt the epidemic [36].

On a personal level, factors leading to *C. trachomatis* testing could be demands from a partner to assure disease free status, or entering a new relationship were both partners tested before engaging in unprotected sex. For men who were not in a stable relationship, testing was done due to uncertainty regarding concurrent sexual relationships of one's partner. Similar reasoning behind testing has previously been described in the same setting [19]. Testing was done to guarantee "cleanliness" and rather than asking a sexual partner about sexual exclusiveness in the relationship, young men and women chose to compensate lack of trust in a partner by testing [13].

Further aspects fostering testing were good knowledge on *C. trachomatis* asymptomatic nature, knowledge on where to test and previous positive experiences of going to a clinic. In a study among students in the US [37], authors found a positive correlation between knowledge and self-efficacy, and also between self-efficacy and STI-testing. Self-efficacy was defined as the self-rated ability to perform a certain behaviour i.e. get tested within the next 6 months [37]. These results indicate that testing coverage can be increased by increasing self-efficacy [37]. The proactive discourse found in our study might translate to high self-efficacy and consequently they test regardless of symptoms and risk-exposure.

The facilitating factors lead to increased testing, facilitating a person´s transition from one discourse to another, i.e., from the indifferent to the ambivalent, to the proactive discourse. It is indeed important that young men exposed to STI go to test. It is known that the level of

infections are higher among male testers compared to female tester [13], and that men can be considered drivers of the infection [33].

However, the proactive discourse towards testing found in our study, might also result in overuse of the health care resources without decreasing the STI prevalence in the country. One example of this is when testing is used instead of preventive strategies such as condom use as it has previously reported in this setting [19]. Testing as means of prevention might lead to people re-infecting themselves if their risk sexual behaviours are not changed exposing themselves and others not only to *C. trachomatis*, but to other more severe STIs such as HIV, syphilis and gonorrhoea among others. Furthermore, testing without changing behaviour can be perceived as wasteful use of health care resources, as testing and treatment exceeds to cost of condoms. Thus, rather than increasing testing levels for men in general it is important to reach those at risk. For example, a British study [38], revealed that the screening program in the UK failed to reach both young men and women engaging in risky sexual behaviour. In the same setting as the present study, health care providers described the importance of finding a balance between primary prevention (condom promotion) and secondary prevention (testing) [39], and that the encounter at the clinic facilitated finding those at risk, given that sufficient time was dedicated to each individual seeking health care.

## Gendered perception of health seeking behaviour

In our study, men described gender differences in how testing for STIs is framed in society influenced their testing behaviour. They highlighted how it has been normalized for women but not for men and how this translated into more health system opportunities for testing. Our findings are in line with studies conducted in Sweden [40] and Canada [41] and highlight the need to continue to devise more strategies to foster men´s use of health services for testing.

Even tough men in our study who previously visited a clinic for STI-testing were satisfied with the reception at the clinic, the fact that the clinics were viewed as a place for women, this can be considered a constraining factor for STI testing. Female friendly referred to as stated above, increasing male testing could halter the *C. trachomatis* epidemic, nevertheless women do face more sexual and reproductive health issues related to for instance menstrual problem, contraception prescription, genital pain caused by Candida infection; thus, it is natural that clinics are female friendly and that women are offered to test more frequently. The target ratio for male versus female testing should be discussed, as well as reasons for testing.

Among the constraining factors for *C. trachomatis* testing, we found knowledge gap regarding the actual testing procedure. Men expresses an uncertainty around the urethral test being painful and whether one would need to undress and undergo a physical examination of the genitals which is in line with the findings of one study conducted Vancouver, Canada [41]. Thus, creating male friendly clinics and further inform about the simplicity of a *C. trachomatis* test might be a way forward to increase male testing. Furthermore, some men indicated that they were afraid of being judged by the staff at the clinics, yet another reason to adjust clinics to suit both men and women.

## Social support: Peers as facilitators to testing

Testing facilitated by social support from peers was a key finding in this study. For the indifferent and the ambivalent discourse, the social support can play an important role in promotion of testing. Young men in our study expressed helpful measures, such as attending the clinic together with a friend or getting information about the testing procedure from a peer, as crucial. Social support from peers is known to expedite testing, although many studies we conducted with the focus of HIV-testing among men who have sex with men (MSM) [42]. In this

population, a meta-analysis showed that peer-led interventions increase testing rates for HIV among MSM [43].

Heterosexual young men and women in a Dutch study [44], highlighted the importance of disclosing testing to a close circle of peers, as well as encouraging/promoting testing within the same group. Thus, to avoid anticipated stigma related to testing, these youth kept their testing behaviour within a trusted peer network [44]. These finding correspond to our study as young men did not only reflect upon the supportive role, but also the judgemental and hindering role of peers in relation to sexual health and testing. Men in our study had previous experiences of peers laughing and joking about *C. trachomatis* and STI and they were afraid of being judged, which is why they often choose not to disclose testing or STI status. Likewise, Cassidy (2019) [45], and Knight (2012) [46] found that peer influenced sexual health both in terms of facilitating and hindering factors for sexual health service accessibility. These findings, as well as ours, indicate the need to foster intervention to harness the power of peer support to encourage STI testing while also promoting peers´ empathic listening and counselling.

## Masculinities and testing

Although testing for STIs was not considered unmanly by our informants, domains of traditional masculinity such as leaning on self-reliance [47] (reluctance to seek help), beliefs on invulnerability and framing men as more carefree with their sexual health than women hindered some young men from testing for STIs. As discussed by Shoveller et al [41] and Sui et al [48], recognizing the need for help can threaten the self-image of men that associate masculinity with high levels of resilience, independence and invulnerability. The discursive construction of men as carefree and less responsible for their sexual health than women is problematic as it can be used to justify young men´s lack of action on their sexual health needs. Our findings indicate the need to continue challenging hegemonic forms of masculinities that hinder young men´s health.

## Strengths and limitations

One limitation includes in-depth interview method on a sensitive subject which could potentially prevent participants from answering truthfully. The study was conducted in Stockholm, a high income setting with good access to health care facilities. Thus, the findings are only applicable in the same setting. All except one man were heterosexual and so men who have sex with men might have different views on sexual risk taking and testing.

One of the strengths of this study was the high number of informants. Furthermore, trustworthiness was assured by detailed description of the study setting (transferability). Additionally, during the interviews, interpretive questions were used to validate the understanding of the data [29]. Confirmability (neutrality of the findings), was assured by repeat reflection and discussions within the research-team. The findings were triangulated by researchers with different backgrounds.

## Conclusions

*C. trachomatis* can be prevented by dissemination of knowledge, consultancy, behavioral change, as well as testing and treatment. Health systems aiming to increase testing among those at risk should take into consideration the positive role that young men's social support have, especially the level of social support coming from their peers. Additionally, endorsement of traditional masculinity values may delay or hinder testing. Promotion of safe sex practices (condom use) should also continue in this setting especially among those at risk.

## Acknowledgments

The authors want to thank all members of the research group GLOSH at Karolinska Institute for their constructive comments and discussions. We also want to thank all the informants who participated in the study.

## Author Contributions

**Conceptualization:** Mariano Salazar.

**Formal analysis:** Frida M. Larsson, Erica Briones-Vozmediano, Johanna Stjärnfeldt, Mariano Salazar.

**Funding acquisition:** Mariano Salazar.

**Methodology:** Mariano Salazar.

**Project administration:** Frida M. Larsson, Mariano Salazar.

**Supervision:** Mariano Salazar.

**Writing – original draft:** Frida M. Larsson, Anna Nielsen, Mariano Salazar.

**Writing – review & editing:** Frida M. Larsson, Anna Nielsen, Erica Briones-Vozmediano, Johanna Stjärnfeldt, Mariano Salazar.

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
