## [Decision Letter · Decision Letter 0]

13 Jul 2021

PONE-D-21-05813

Indifferent, ambiguous, or proactive? Young men’s discourses on health service utilization for Chlamydia detection in Stockholm Sweden; a qualitative study.

PLOS ONE

Dear Dr. Larsson,

Thank you for submitting your manuscript to PLOS ONE. After careful consideration, we feel that it has merit but does not fully meet PLOS ONE’s publication criteria as it currently stands. Therefore, we invite you to submit a revised version of the manuscript that addresses the points raised during the review process.

I apologize for the very substantial delay in handling your submission. I became involved as an editor only moe recently. I have read your manuscript myself, and it has been seen and assessed by an outside specialist reviewer. The reviewer makes only few, more editorial, comments, which are easily satisfied (the spelling should indeed by *Chlamydia trachomatis*, but CT is fine as an abbreviation).

This is not really my field of scientific activity and I realize that you write according to the standard in the field but would it not be possible to give some sort of semi-quantitative assessment of what share of the participants fall in which group (ambiguous etc.)? It may not be representative of the population ut would still give information.

Finally, I am not sure that, as you say in the abstract, you can conclude from the study that testing cannot replace safe sex practices. There is no doubt that the statement is correct, but from which of your findings would you conclude that?

We look forward to receiving your revised manuscript.

Kind regards,

Georg Häcker

Academic Editor

PLOS ONE

Journal Requirements:

1. Please ensure that your manuscript meets PLOS ONE's style requirements, including those for file naming. The PLOS ONE style templates can be found athttps://journals.plos.org/plosone/s/file?id=wjVg/PLOSOne_formatting_sample_main_body.pdf and https://journals.plos.org/plosone/s/file?id=ba62/PLOSOne_formatting_sample_title_authors_affiliations.pdf

Additional Editor Comments (if provided):

Reviewers' comments:

Reviewer's Responses to Questions

**Comments to the Author**

1. Is the manuscript technically sound, and do the data support the conclusions?

Reviewer #1: Yes

2. Has the statistical analysis been performed appropriately and rigorously? 

Reviewer #1: N/A

3. Have the authors made all data underlying the findings in their manuscript fully available?

Reviewer #1: Yes

4. Is the manuscript presented in an intelligible fashion and written in standard English?

Reviewer #1: Yes

5. Review Comments to the Author

Reviewer #1: I have read with interest this paper. I believe that the paper is interesting. However, I have some concerns that are reported herein.

The title can be improved for better understanding.

Please change Chlamydia Trachomatis to Chlamydia trachomatis in italic form; it is a bacterium anyway.

Please avoid using CT to represent C. trachomatis, use this form instead.

Please check the following errors or mistakes:

“Most reported gymnasium as the highest level of education (14/18)” This is so confusing.

“Thus, CT was constructed as a mild curable disease when compared other STI: s such as HIV” Is other STI: s acceptable? Similar words are used in the text. Please correct them.

Change candida infection to Candida infection.

“why it is natural that clinics are female friendly and that women are offered to test more frequently” is “why” correctly used here?

6. PLOS authors have the option to publish the peer review history of their article (what does this mean?). If published, this will include your full peer review and any attached files.

Reviewer #1: No

---

## [Author Response · Author response to Decision Letter 0]

20 Aug 2021

Dear editor, 

We thank you and the reviewers for your very useful comments and notification regarding our Data Availability statement. In the following, we will give a detailed answer to the queries made regarding our Data Availability statement. 

1. PLOS journals require that all data presented in the study be made publicly available at or before the time of publication. If there are legal or ethical restrictions on the data being made publicly available, such as IRB restriction or patient confidentiality, authors must provide a way for fellow researchers to access the data. Please note that PLOS does not allow authors to be the sole contact for data inquiries. If the data is only available upon request, please provide contact information, such as an email address, for a non-author, institutional point of contact (such as an IRB or ethics committee contact) who can field data inquiries from fellow researchers. If the data contact is an individual, please provide their title and relationship to the data as well. 

A. We have changed the Data Availability statement to: “Our paper contains qualitative data. Following Plos One guidelines, excerpts of the data transcripts that are relevant for the study are available within the paper in the result section. Requests for access to more data should be made to the Research Data Office at Karolinska Institutet via rdo@ki.se and if permitted by law and ethical approval, decided on a case by case basis, the data can shared.”

For your information, the previous comments from the editor and reviewers are also described below. 

All the best, 

Frida Larsson

Journal Requirements

In your revised cover letter, please address the following prompts :a) If there are ethical or legal restrictions on sharing a de-identified data set, please explain them in detail (e.g., data contain potentially identifying or sensitive patient information) and who has imposed them (e.g., an ethics committee). Please also provide contact information for a data access committee, ethics committee, or other institutional body to which data requests may be sent.

A: Thank you for this question. Our paper contains qualitative data. We follow the EU General Data Protection Regulation (2016/679) regarding restricted access to the data. The study protocol was approved by Stockholm Ethic Review Board (https://www.registerforskning.se/en/the-ethics-review-boards-become-the-swedish-ethical-review-authority/). The sensitive nature of our qualitative data means that even if the names and other identifying information (address, telephone number, etc.) of the young men part of this research have been removed from the transcripts, there is a possibility that they could be identified thorough their narratives (as they share personal experiences). Thus, any request for access must be approved by the project Principal investigator Mariano Salazar: mariano.salazar@ki.se and Karolinska Institutet legal team.

Editor´s comment

1. This is not really my field of scientific activity and I realize that you write according to the standard in the field but would it not be possible to give some sort of semi-quantitative assessment of what share of the participants fall in which group (ambiguous etc.)? It may not be representative of the population ut would still give information.

A: Thank you for this question. In qualitative research, the focus is to understand, describe or explain a phenomenon not to quantify it [1]. We agree that it will great to quantify the frequency of the discourses identified in our study in the study population but a better method to do this is a study using a quantitative approach. 

2. Finally, I am not sure that, as you say in the abstract, you can conclude from the study that testing cannot replace safe sex practices. There is no doubt that the statement is correct, but from which of your findings would you conclude that?

A: Thank you for you comment. We have revised the abstract according to your suggestion and made other small changes. The abstract reads as follows:

“Introduction. Chlamydia trachomatis (C. trachomatis) infection is the most commonly reported sexually transmitted infection in Sweden and globally. C. trachomatis is often asymptomatic and if left untreated, could cause severe reproductive health issues. In Sweden, men test for C. trachomatis to a lesser extent than women. 

Aim. To explore factors facilitating and constraining Swedish young men’s health care utilization for C. trachomatis detection and treatment. 

Method. A qualitative situational analysis study including data from 18 semi-structured interviews with men (21-30 years). Data collection took place in Stockholm County during 2018. A situational map was constructed to articulate the positions taken in the data within two continuums of variation representing men’s risk perception and strategies to test for C. trachomatis.

Results. Based on the informants’ risk perception, strategies adopted to test and the role of social support, three different discourses and behaviours towards C. trachomatis testing were identified ranging from a) being indifferent about C. trachomatis -not testing, b) being ambivalent towards testing, to c) being proactive and testing regularly to assure disease free status. Several factors influenced young men’s health care utilization for C. trachomatis detection, where the role of health services and the social support emerged as important factors to facilitate C. trachomatis testing for young men. In addition, endorsing traditional masculinity domains such as leaning on self-reliance, beliefs on invulnerability and framing men as more carefree with their sexual health than women delayed or hindered testing.

Conclusion. Testing must be promoted among those young men with indifferent or ambivalent discourses. Health systems aiming to increase testing among those at risk should take into consideration the positive role that young men’s social support have, especially the level of social support coming from their peers. Additionally, endorsement of traditional masculinity values may delay or hinder testing.”

Reviewer #1: I have read with interest this paper. I believe that the paper is interesting. However, I have some concerns that are reported herein.

1. The title can be improved for better understanding. Please change Chlamydia Trachomatis to Chlamydia trachomatis in italic form; it is a bacterium anyway.

A: Thank you for this comment. The title has changed according to your suggestion. It reads as follows: “Indifferent, ambiguous, or proactive? Young men’s discourses on health service utilization for Chlamydia trachomatis detection in Stockholm, Sweden; a qualitative study.”

2. Please avoid using CT to represent C. trachomatis, use this form instead.

A: Thank you for this comment. We have changed the wording throughout the manuscript. 

3. Please check the following errors or mistakes: “Most reported gymnasium as the highest level of education (14/18)” This is so confusing.

A: Thank you for this. In Sweden gymnasium refers to upper secondary school. We have changed the sentence to reflect this. The text reads as follows (page 7, 3rd paragraph): “Most reported upper secondary school as their highest level of education (14 of 18 young men)” 

4. “Thus, CT was constructed as a mild curable disease when compared other STI: s such as HIV” Is other STI: s acceptable? Similar words are used in the text. Please correct them.

A: Thank you for this comment. We have revised the text. It reads as follows (page 10, 1st paragraph): “Thus, C. trachomatis was constructed as a mild curable disease when compared other STIs such as HIV.”

5. Change candida infection to Candida infection.

A: Thank you. The change has been made. The text reads as follows (page 20, 1st paragraph): “genital pain caused by Candida infection”. 

6“why it is natural that clinics are female friendly and that women are offered to test more frequently” is “why” correctly used here?

A: Thank you. We have changed the sentence to (page 20, 1st paragraph): “genital pain caused by Candida infection; thus, it is natural that clinics are female friendly and that women are offered to test more frequently.” 

References: 

1. Austin Z, Sutton J: Qualitative research: getting started. Can J Hosp Pharm 2014, 67(6):436-440.

Journal Requirements:

1-Please ensure that your manuscript meets PLOS ONE's style requirements, including those for file naming. The PLOS ONE style templates can be found at https://journals.plos.org/plosone/s/file?id=wjVg/PLOSOne_formatting_sample_main_body.pdf and https://journals.plos.org/plosone/s/file?id=ba62/PLOSOne_formatting_sample_title_authors_affiliations.pdf

---

## [Editor Report · Decision Letter 1]

1 Sep 2021

Indifferent, ambiguous, or proactive? Young men’s discourses on health service utilization for Chlamydia trachomatis detection in Stockholm, Sweden; a qualitative study.

PONE-D-21-05813R1

Dear Dr. Larsson,

We’re pleased to inform you that your manuscript has been judged scientifically suitable for publication and will be formally accepted for publication once it meets all outstanding technical requirements.

Kind regards,

Georg Häcker

Academic Editor

PLOS ONE
---

## [Editor Report · Acceptance letter]

23 Sep 2021

PONE-D-21-05813R1 

Indifferent, ambiguous, or proactive? Young men’s discourses on health service utilization for *Chlamydia trachomatis* detection in Stockholm, Sweden; a qualitative study. 

Dear Dr. Larsson:

I'm pleased to inform you that your manuscript has been deemed suitable for publication in PLOS ONE. Congratulations! Your manuscript is now with our production department. 

Kind regards, 

on behalf of

Dr. Georg Häcker 

Academic Editor

PLOS ONE